# Noncompetitive Chromogenic Lateral-Flow Immunoassay for Simultaneous Detection of Microcystins and Nodularin

**DOI:** 10.3390/bios9020079

**Published:** 2019-06-18

**Authors:** Sultana Akter, Teemu Kustila, Janne Leivo, Gangatharan Muralitharan, Markus Vehniäinen, Urpo Lamminmäki

**Affiliations:** 1Molecular Biotechnology and Diagnostics, Department of Biochemistry, University of Turku, FIN-20520 Turku, Finland; Teemu.kustila@gmail.com (T.K.); jpleiv@utu.fi (J.L.); drgm@bdu.ac.in (G.M.); marvehni@utu.fi (M.V.); urplammi@utu.fi (U.L.); 2Department of Microbiology, School of Life Sciences, Bharathidasan University, Palkalaiperur, Tiruchirappalli 620024, Tamilnadu, India

**Keywords:** noncompetitive immunoassay, cyanotoxin, microcystin, nodularin, lateral flow

## Abstract

Cyanobacterial blooms cause local and global health issues by contaminating surface waters. Microcystins and nodularins are cyclic cyanobacterial peptide toxins comprising numerous natural variants. Most of them are potent hepatotoxins, tumor promoters, and at least microcystin-LR is possibly carcinogenic. In drinking water, the World Health Organization (WHO) recommended the provisional guideline value of 1 µg/L for microcystin-LR. For water used for recreational activity, the guidance values for microcystin concentration varies mostly between 4–25 µg/L in different countries. Current immunoassays or lateral flow strips for microcystin/nodularin are based on indirect competitive method, which are generally more prone to sample interference and sometimes hard to interpret compared to two-site immunoassays. Simple, sensitive, and easy to interpret user-friendly methods for first line screening of microcystin/nodularin near water sources are needed for assessment of water quality and safety. We describe the development of a two-site sandwich format lateral-flow assay for the rapid detection of microcystins and nodularin-R. A unique antibody fragment capable of broadly recognizing immunocomplexes consisting of a capture antibody bound to microcystins/nodularin-R was used to develop the simple lateral flow immunoassay. The assay can visually detect the major hepatotoxins (microcystin-LR, -dmLR, -RR, -dmRR, -YR, -LY, -LF -LW, and nodularin-R) at and below the concentration of 4 µg/L. The signal is directly proportional to the concentration of the respective toxin, and the use of alkaline phosphatase activity offers a cost efficient alternative by eliminating the need of toxin conjugates or other labeling system. The easy to interpret assay has the potential to serve as a microcystins/nodularin screening tool for those involved in water quality monitoring such as municipal authorities, researchers, as well as general public concerned of bathing water quality.

## 1. Introduction

Blooms of cyanobacteria, a phylum of bacteria that obtain their energy through photosynthesis, are commonly found around the globe. Certain cyanobacterial strains are known to produce toxic secondary metabolites called cyanotoxins, which cause diverse problems by contaminating water sources used for recreation, drinking water production and (aqua) farming. Generally, the most widespread and problematic cyanotoxin group consists of cyclic peptide toxins comprising numerous natural variants. Close to 250 analogues of microcystin and about 10 analogues of nodularin with differing toxicity have been reported to date in the literature [1]. The heptapeptide, microcystin in fresh water, and pentapeptide nodularin in brackish water bodies are the potent hepatotoxins acting as specific inhibitors of protein phosphatases (PPs) and thus hazardous to health, being responsible for growth of tumor precursors, particularly in liver [2,3]. Cyanotoxins have the potential to accumulate in fish and marine organisms via food chains [4]. Microcystin and nodularin are very stable and resistant to hydrolysis, making them a persistent issue for common water usage [5]. The robustness results from the chemical structure of microcystin and nodularin, which consists of a monocyclic peptide ring and a hydrophobic Adda moiety (3-amino-9-methoxy-2,6,8-trimethyl-10-phenyldeca-4(*E*),6(*E*)-dienoic acid). The conserved Adda group is the common determinant for all microcystin and nodularin, and has proven useful in the development of broad-specificity detection methods [6,7]. Microcystin-LR is the widely distributed and most studied microcystin analogue. Similarly, nodularin-R is the most common nodularin variant found in the brackish water bodies such as in the Baltic sea. World Health Organization (WHO) has established guidelines for cyanotoxins in the assessment of drinking water quality and recreational use [8,9]. For drinking water, WHO set guideline value is 1 µg/L of microcystin-LR. In addition, several countries have issued guideline values for recreational water use where the microcystin-LR levels usually range between 4.0–25 µg/L [10].

The large number of toxin variants combined with the low-molecular weight of microcystins and nodularins (~1000 Da) creates major challenges for the development of efficient analytical detection methods. The confirmatory detection and identification of cyanotoxins is routinely done in centralized laboratories with the use of high-sensitivity reference methods such as liquid chromatography (LC) [11,12], mass-spectrometry (MS) [13], or a combination of both [14,15]. Although these methods are well suited for the exact identification of the contaminating toxin, the need for specialized instrumentation and trained staff to operate them makes them unsuitable for rapid or high-throughput testing. Alternative methods for cyanotoxin testing have also been established, particularly methods based on protein phosphatase inhibition (PPI) [16,17] and antibodies [6,18] are widely used due to the simplicity and cost-efficiency of the assays. Although these methods are widely used in commercial kits, both techniques have certain limitations. Usually, these methods are performed in laboratory and are not suited easily for on-site use.

One of the main benefits of using antibodies for cyanotoxin detection is the possibility to implement the assay format for field testing, as the result of the possible contamination is often needed rapidly, and the guideline values do not require exceptional sensitivity. Among such methods, lateral-flow immunoassays (LFIA) have gained considerable interest in the academic and industrial research [19,20]. LFIA immunostrips are widely used to target various biomolecules in a myriad of commercial applications, ranging from monitoring of drug abuse to environmental contaminants [21]. Lateral-flow assays provide rapid results in a very cost effective and de-centralized manner. The results for LFIA can be observed without the need of any special expertise and thus it offers wide applicability for testing location. Field testing is particularly useful for cyanotoxin detection, as easy monitoring tools for surface waters at multiple sites are needed to meet the established guideline values. However, the simultaneous and simple detection of all the different variants is extremely challenging and the low-molecular weight of cyanotoxins has limited the available detection methods used in LFIA to competitive assay formats [22,23]. Furthermore, competitive immunoassays are intrinsically more prone to sample interference and the results are hard to interpret compared to the noncompetitive counterparts. In addition, most of the currently available LFIA assays utilize conjugated toxin derivatives as a reporter system for the assay, which are often expensive to manufacture.

We describe the development of a simple chromogenic LFIA for simultaneous detection of microcystins and nodularin. The assay utilizes recombinant single-chain antibody fragment-alkaline phosphatase (scFv-AP) fusion protein as a reporter, which provides a readymade tracer system where the signal intensity on the immunostrip test line is directly proportional to the toxin concentration. Moreover, the result can be visually confirmed without the need for separate reading device. The sensitivity of the LFIA was found to be well below the 4 µg/L of microcystin-LR equivalents, the minimum guideline value set for recreational water use by certain countries such as Hungary. The LFIA was capable of simultaneous detection of microcystins and nodularin-R from environmental water samples collected at sites known to be contaminated with cyanotoxins during the cyanobacterial blooming season. The assay can be used as a screening tool for the detection of microcystin and nodularin from surface waters.

## 2. Materials and Methods

### 2.1. Common Materials and Reagents

Common inorganic and organic chemical reagents were obtained from commercial source Sigma or Merck unless otherwise specified. The reagent water used was purified by Millipore Milli-Q Plus water filtration purification system (Millipore Corporation, Burlington, MA, USA). Enhancement solution, wash concentrate, and streptavidin coated microtiter plates were from Kaivogen (Turku, Finland). Monoclonal anti-Adda antibody, AD4G2 (Adda specific, anti-Microcystins) was from Enzo Life Sciences, Inc. (Farmingdale, NY, USA). Bacterial anti alkaline phosphatase polyclonal antibody (anti-AP Pab) which was purchased from LifeSpan Biosciences, Inc. (Seattle, WA, USA) was purified in house through protein G affinity purification. Histidin tag scFv purification was done by His Spin Trap™ kit (GE Healthcare, Amersham, UK). The bacterial host *Escherichia coli* XL-1 Blue was from Stratagene, La Jolla, CA. Fc specific monoclonal human anti-mouse IgG (HAMA) which recognize mouse IgG via the Fc region was a gift from Dr. Keith Thompson (University of Oslo). Alkaline phosphatase substrate BCIP/NBT (5-Bromo-4-chloro-3-indolyl phosphate BCIP and nitro blue tetrazolium NBT) tablets were purchased from SIGMA. According to the manufacturer’s instruction, one tablet was dissolved in 10 mL of water yielding substrate solution of BCIP (0.15 mg/mL), NBT (0.30 mg/mL), Tris buffer (100 mM), and MgCl_2_ (5 mM), pH 9.25–9.75. Lateral flow assay buffer (LFAB) was composed of 10 mM Phosphate, 137 mM NaCl, 2.7 mM KCl, pH 7.3; supplemented with 0.5% Tween-20, 1% BSA, 0.06% bovine γ-albumin, and filtered through a 0.22 µm filter. Once prepared, it was kept at 4 °C and used for two weeks. Three times LFAB (3 × LFAB) was prepared using the above composition with three times molar excess. Superb broth (SB medium, pH 7) was composed of 2% yeast extract, 3% tryptone, and 1% MOPS.

### 2.2. Instrumentation

Multilabel counter Victor^TM^ 1420 for fluorescence measurement was from PerkinElmer Life Sciences, Finland. Protein concentration were measured by NanoDrop ND1000 spectrophotometer (Thermo Scienctific, Waltham, MA, USA). A Linomat 5 sample applicator (CAMAG, Muttenz, Switzerland) was used for striping of the binder and control line molecule. A desktop paper cutter (Ideal 1058, Krug & Priester, Balingen, Germany) was used to cut the test strips.

### 2.3. Toxin Standards

Specific amount of the purified toxins were obtained from Dr. Jussi Meriluoto’s Lab (Åbo Akademi University) as a lyophilized dried powder (microcystin-LR, -dmLR, -RR, -dmRR, -YR, -LY, -LF -LW, nodularin-R, and anatoxin-a) or as solution (cylindrospermopsin). The microcystins and nodularins were purified as described previously [24]. Dry powder of microcystin and nodularin was dissolved in 50% methanol yielding 100 to 250 µM original stock. Dried anatoxin-a was dissolved in reagent water (~10 µg/mL original stock). Further working standard stocks of all toxins were diluted in reagent water and kept at −20 °C for long term or at 4 °C for short term in sealed condition.

### 2.4. Anti-Immunocomplex Antibody Fragment

The generic anti-immunocomplex (anti-IC) single-chain fragment (scFv) SA51D1 as fusion to alkaline phosphatase (scFv-AP SA51D1) reported in Akter et al., 2016 [25] was used in this work to develop the non-competitive sandwich-type LFIA. The isolation, purification and characterization of the anti-IC scFv-AP has been described in detail in Akter et al., 2016 [25]. The scFv-AP was expressed in XL-1 Blue *E. coli* cells in 50 mL culture in SB medium supplemented with 100 µg/mL ampicillin, 10 µg/mL tetracycline, 0.05% glucose, and induced at 26 °C for 4–6 h. Harvested cells were purified through histidin tagged scFv-AP using His trap affinity column (GE Healthcare) according to the manufacturer’s instructions. In Akter et al., 2016, [25] we reported the use of the anti-IC scFv-AP to develop a highly sensitive time-resolved fluoroscence based IC assay (TRF-IC assay) capable of detecting all the tested 11 different cyanobacterial peptide hepatotoxin (microcystin-LR, -dmLR, -RR, -dmRR, -WR, -YR, -LA, -LY, -LF, -LW, and nodularin-R) well below WHO guide line limit of 1 µg/L. The scFv-AP does not have any significant binding affinity towards naked anti-Adda Mab nor to the toxin alone [25]. Furthermore, using the scFv-AP, we also reported a non-competitive ELISA with broad specificity for microcystin and nodularin utilizing the fusion AP enzyme to generate visually detectable signal [26].

### 2.5. Preparation of Immunostrips

The lateral-flow immunostrip (Figure 1) used to perform the LFIA consists of cellulose absorption pad (Millipore, USA), nitrocellulose membrane (Hi-Flow Plus HF180, Millipore, USA), and glass fiber feeding pad (Millipore, USA).

A 2.5 cm wide nitrocellulose membrane was attached to the plastic adhesive backing card (G&L Precision Die Cutting, San Jose, CA, USA). An adsorption pad was prepared by attaching a 34 mm wide cellulose membrane strip (Millipore, MA, USA) to overlap 2–3 mm with the nitrocellulose membrane. A feeding pad of 16 mm wide glass fiber strip (Millipore, MA, USA) was attached to overlap 2–3 mm with the other end of the nitrocellulose membrane.

The striping/printing of the test line was performed by dispensing 0.5 mg/mL of HAMA in 10 mM Tris–HCl buffer (pH 8.0) with 1% methanol on the nitrocellulose membrane. The control line was printed similarly at a 5-mm distance by dispensing 0.6 mg/mL anti-AP Pab in the same buffer composition. Printing was accomplished using Linomat 5 sample applicator (CAMAG, Muttenz, Switzerland) which was adjusted to produce 1µL/cm stripes with a liquid flow speed of 250 nL/s. Membranes were then dried at 37 °C for 2–3 h. The assembled LFIA cards were then cut into 4 or 5 mm wide chips/strips using a desktop paper cutter (Ideal 1058, Krug & Priester, Balingen, Germany). In each experiment, same-width strips were used.

### 2.6. Noncompetitive Sandwich-Type LFIA Procedure

The LFIA was performed in a standard 96 well polypropylene microtiter plate (Greiner BioOne, Kremsmünster, Austria) and consisted of four steps (Figure 1b). (1) A pre-incubation step: 40 µL of reagent water (as blank)/sample/toxin standard was added to a 20 µL of reaction mix (20 ng anti-Adda Mab and 35 ng anti-IC scFv-AP SA51D01 in 3 × LFAB) and incubated 10–15 min at slow shaking. (2) Feeding step: In this step the feeding pad of the immunostrips were dipped in the total 60 µL reaction volume. The absorption of the liquid took place within approximately 10 min. (3) Washing step: In the washing step, stripes were moved into a well containing 100 µL of LFAB and left for 25 min. (4) Colorimetric reaction: In this fourth step, the strips were moved to the reaction well containing 150 µL of BCIP/NBT substrate solution. The absorption and colorimetric reaction was completed within about 2 h. All steps were carried out in room temperature (RT). The detailed principle and procedure of the chromogenic LFIA is described in Figure 1.

### 2.7. Applicability of the LFIA in the Detection of Different Hepatotoxins

Nine different common cyanobacterial hepatotoxins (heptapeptide microcystin-LR, -dmLR, -RR, -dmRR, -YR, -LY, -LF, -LW, and pentapeptide nodularin-R) were used to assess the specificity of the LFIA. The toxins were spiked in to LFAB at the concentration of 4 µg/L. The LFIA procedure was carried out as described earlier in Section 2.6.

### 2.8. Detection of Microcystin-LR from Spiked Water Samples

Reagent water and two raw environmental surface water samples from Finnish lake (S11: Alasenjärvi, Lahti, 6 July, 2009 and V1: Rusutjärvi, 23 June, 2009) were spiked with microcystin-LR over a range of concentrations (0, 1, 4, 10, 20 µg/L). The used environmental samples collected in 2009 were previously analyzed for intracellular toxin content by LC-MS and were stored at −20 °C. Trace amount of cyanobacterial hepatotoxin (0.07 and 0.04 µg/L, respectively) was detected from these two samples. The LFIA was performed using duplicate strips according to procedure described in Section 2.6.

### 2.9. LFIA Performance with Environmental Water Samples

Fourteen raw surface water samples collected during 2009 from Finland and Estonia were analyzed by the LFIA to detect total toxin (extracellular and intracellular). The samples were frozen and thawed at least twice to break the cells and to allow the toxins to be released in the samples. No further extraction or process was performed on the samples. Furthermore, the same raw water samples were analyzed according to the previously reported TRF-IC assay [26]. In addition, for each corresponding samples, intracellular toxin concentration result (using extracted cell samples) by LC-MS was available which was reported earlier by Hautala et al. [27].

### 2.10. LFIA Performance with Possible Interference

Reagent water was spiked with microcystin-LR, anatoxin-a and cylindrospermopsin at concentration ranging 1 to 100 µg/L of toxin. The LFIA was performed using duplicate strips according to procedure described in Section 2.6. Water sample containing no toxin was tested as a control.

### 2.11. Data Analysis and Result Interpretation

The results of the LFIA was done by visual inspection, and the images were recorded using Canon EOS 60D DSLR (Canon Inc., Tokyo, Japan), Samsung Galaxy A5 2016 mobile phone camera (Samsung, Seoul, South Korea) and ChemiDoc MP Imaging System (Bio-Rad, Hercules, CA, USA). The images were processed with Corel Photo-Paint 2018 software (Corel Corporation, Ottawa, ON, Canada). Briefly, the images were cropped and adjusted parallel for illustrative purposes and the images were converted to greyscale.

## 3. Results

### 3.1. LFIA Development and Detection of Microcystin-LR from Spiked Samples

We spiked three different water sources (reagent water and two environmental water samples) with known amount of microcystin-LR to assess the initial performance of the LFIA immunostrips. No significant difference could be observed among the water sources or the replicate strips done (Figure 2). For the test, microcystin-LR was selected due to its wide distribution in environmental samples. All of the four microcystin-LR concentrations used in the test resulted in a clear positive test line (Figure 2). Signal intensity increased proportionally with increasing toxin concentration. In addition, the control line could be detected in all immunostrips.

### 3.2. Applicability of the LFIA in the Detection of Different Hepatotoxins

The specificity of the anti-IC scFv-AP SA51D1 towards different cyanobacterial microcystins was extremely board and as reported previously in Akter et al., the scFv-AP was capable of detecting all of the tested 11 different cyanotoxins (microcystin-LR, -dmLR, -RR, -dmRR, -WR, -YR, -LA, -LY, -LF, -LW, and nodularin-R) [25]. The board specificity profile was confirmed also for the LFIA with the use of nine different cyanotoxins (nodularin-R; and microcystin-LR, -dmLR, -RR, -dmRR, -LF, -LY, -LW, and -YR) as analytes. The LFIA strips (see Appendix A) were used to analyze the 4 µg/L of toxin analyte. All of the analyzed samples gave clear positive test lines (Figure 3).

### 3.3. LFIA Performance with Enviromental Samples

The LFIA performance with natural samples was tested with the use of 14 environmental samples collected from various locations during the blooming season in southwest Finland and Estonia. For each sampling site, two parallel type of samples were available: untreated raw water and filtered cell. The raw waters were analyzed with the developed LFIA as well as with TRF-IC assay [25]. Intracellular toxin content from extracted cells were previously analyzed by LC-MS and was reported by Hautala et al. [27]. The details of the samples are summarized in Table 1.

The LFIA immunostrips results were visually inspected (Figure 4). Two of the 14 samples previously confirmed as negative for all cyanotoxin variants [27] yielded no visible test lines. The positive samples based on reference method yielded faint to intense test lines in LFIA immunostrips depending on the toxin concentration. Two of the sampling locations contained heavy blooming of algae and consequentially had high concentrations of cyanotoxins (>25 µg/L).

### 3.4. LFIA Performance with Possible Interference

The LFIA performance with possible interfering agents are investigated with the use of other cyanobacterial toxin such as anatoxin-a and cylindrospermopsin. As expected, the presence of high amount (100 µg/L) of anatoxin-a and cylindrospermopsin did not result in any false positive or inconclusive decision (Figure 5).

## 4. Discussion

The adverse effects caused by freshwater cyanobacteria are of major concern as some of the metabolites are toxic to humans and other mammals, hamper plant growth, and may accumulate in sea food. The diversity and complexity of the toxins produced by cyanobacteria creates challenges, especially for field testing. Although the reference methods, such as MS (mass spectrometry) and HPLC (high-performance liquid chromatography) for the detection of microcystin and nodularin are well established, more cost-efficient, and rapid methods are urgently needed for field testing. Our goal was to develop a broadly specific noncompetitive LFIA for the simultaneous detection of microcystin and nodularin directly from untreated environmental samples.

The lateral-flow immunostrips were designed to provide a simple and reproducible result with the use of recombinant antibody fragment in fusion with bacterial AP. The use of AP as a reporter on the immunostrip had two clear benefits over previously described LFIA systems. (1) The scFv-AP as a chromogenic reporter in the LFIA enabled the simple design of the immunostrip, where the readymade tracer can be produced with simple bacterial expression. This removes the need for any chemical conjugation steps of additional reporter molecules, and at the same time, keeping the molar ratio of the antibody-label constant. (2) The excess scFv-AP not bound to the immunocomplex of the anti-Adda Mab and the cyanotoxin was used to establish the control line for the assay. The use of HAMA instead of the Adda Mab on the test line was done to ensure the formation of the IC before the lateral-flow step. Since the formation of the IC is dependent on the affinities of the parental Adda Mab to the cyanotoxin, and the scFv, to the IC, the preincubation time was used to gain higher sensitivity in the system. The substrate BCIP/NBT has previously been successfully implemented for paper based immunoassays [28,29]. Different AP substrates were tested for the LFIA during this study (result not shown), and the benefits of BCIP/NBT were confirmed to be superior in comparison to other substrates. As the reaction end product precipitates due to the AP activity, it restricts the diffusion of the signal outside the detection lines. In addition, the reaction end product has a very intense blue-purple color, which forms a good contrast against white nitrocellulose LF-matrix. Moreover, the product is stable and not light sensitive, making it suitable also for field testing.

The immunocomplex formed by the anti-Adda Mab and the toxin was detected with the use of previously characterized antibody-fragment, which is known to bind the formed immunocomplex with uniquely broad specificity for microcystin and nodularin. The possibility for a simultaneous detection of multiple cyanotoxins from different toxin class in a noncompetitive format is ideal for high-throughput testing of samples where the contaminating cyanotoxin is unknown. Our aim was to develop this broad-specificity assay concept in to even simpler format, where the result could be obtained rapidly without the need for any specialized instrumentation. The amounts of anti-Adda Mab (20 ng) and the generic anti-IC scFv-AP (35 ng) were optimized to generate strong enough signals for visual inspection on the test and control lines. Next, we tested the performance of the LFIA by spiking variable amounts of microcystin-LR toxin to reagent water as well as environmental waters. As shown in Figure 2, the increasing amount of microcystin-LR correlated strongly with the signal intensity observed on the test line. This contrasts with currently available lateral flow tests for hepatotoxins, where the test line slowly fades with increasing toxin concentration. Although the signal intensity decreases on the test line with lower microcystin-LR concentrations (<1 µg/L), the positive result can still be confirmed from samples containing as low as 0.7 µg/L of cyanotoxins based on the reference methods (LC-MS).

The previously reported TRF-IC immunoassays [25] have shown that, in addition to the high sensitivity of the assay, it also performs with a very broad specificity to hepatotoxins containing the Adda moiety. We wanted to confirm this result also with the use of LFIA by testing the cross-reactivity of the assay using eight different microcystin congeners, nodularin-R, and a mixture of the all nine different hepatotoxins. The concentration of each toxin was set to 4 µg/L, and all the tested toxin samples contributed clear positive results (Figure 3). This confirms the broad-specificity performance of the IC-assay concept, also with simpler assay concept.

The final step for the LFIA development was to test the performance of the immunostrips with the use of environmental samples. The use of real samples where the toxin concentration is unknown can create problems for the accurate detection of cyanotoxins mainly resulting from cross-reacting compounds causing interferences and unspecific matrix effects. In addition, some of the cyanobacterial strains can produce multiple classes of toxins simultaneously. We tested the LFIA in the presence of anatoxin-a and cylindrospermopsin at concentration range from 1–100 µg/L and no interference is observed (Figure 5). We used 14 samples collected from various locations in Finland and Estonia during the cyanobacterial blooming season (June–September). The sampling locations consisted of both fresh and brackish water areas where cyanotoxins are known to be present. The raw waters used for the LFIA were also analyzed by the previously reported TRF-IC assay [25]. Furthermore, all of the samples were previously analyzed [27] with LC-MS methods for the intracellular toxin content, and the detailed description of sampling locations are described in Table 1. All the samples containing cyanotoxin levels > 4µg/L produced a clear positive test line on the immunostrips (Figure 4). For samples containing toxin concentrations ranging between 0.7–4 µg/L depending on the analytical approach (Å22, RN26, and RN30), the bands are faint but observable. No false positive or false negative was observed by the LFIA. Overall, the test line results for LFIA immunostrips correlate well with the previously found cyanotoxin concentration obtained with confirmatory methods. In future, the LFIA described in this study can be further optimized and implemented also for the detection of cyanobacterial contamination of drinking water, which is a major concern in countries with inadequate water treatment facilities.

## 5. Conclusions

An easy to interpret noncompetitive LFIA test for microcystins and nodularin-R was successfully developed. The assay can detect wide variety of cyanobacterial hepatotoxin well below 4 µg/L which is the lowest detection value used for recreational water in many countries. The LFIA is suitable for field conditions and the result can be visually confirmed with no need for a separate measuring device for result interpretation. The signal intensity is directly proportional to the concentration of toxin. Based on conventional AP activity, the assay offers a cost-effective method for first line screening of microcystins and nodularin for assessment of water quality and safety.

## Figures and Tables

**Figure 1 biosensors-09-00079-f001:**
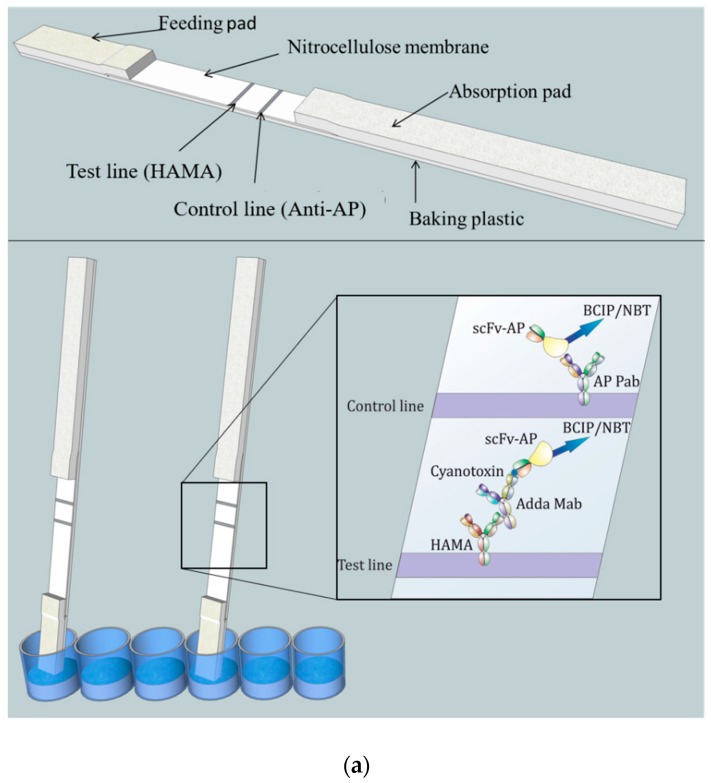
(**a**) Conceptual illustration of the noncompetitive chromogenic LFIA. Top: The structure of the LFIA strip comprising feeding pad, nitrocellulose membrane, and absorbtion pad. Test line (HAMA: human anti-mouse antibody) and control line (anti-AP Pab) were printed on nitrocellulose membrane. Bottom: Principle of the chromogenic anti-IC LFIA. The sample absorbing procedure was performed on microtiter well. Once the liquid is absorbed, the reagents move along the immunostrip by capillary action and respective antibody components attach to the test and control lines. In the presence of MC/Nod in the sample the anti-Adda Mab:MC/Nod:anti-IC-scFv-AP complex become captured by HAMA on the test line through the Fc portion of the anti-Adda Mab. The excess unbound anti-IC-scFv-AP continue to migrate until captured by the anti-AP Pab in the control line. The alkaline phosphatase (AP) fused with the anti-IC scFv triggers an enzymatic reaction producing a chromogenic precipitant in the presence of BCIP/NBT substrate. In the absence of MC/Nod, anti-IC scFv-AP does not bind to the test line and continue to migrate until captured in the control line. The control line ensures that the chromogenic reaction is functional in the LFIA. (**b**) The main steps of the LFIA procedure where the sample absorption took place on microtiter well. The assay protocol comprised of four main steps. (1) In the preincubation step, sample and reagents are mixed together to allow formation of the anti-Adda Mab:MC/Nod:anti-IC-scFv-AP complex in the presence of toxin. (2) In the feeding step, LF chips were dipped into the pre-incubated reaction mixture through feeding pad where antibody components or any formed immunocomplex migrated through test and control line. (3) The washing step removes any unbound antibody components. (4) In the fourth step, colorimetric reaction took place in the presence of chromogenic substrate. (5) Finally, the developed color on the strips can be observed by naked eye or through instrument (optional). MC = microcystin; Nod = nodularin.

**Figure 2 biosensors-09-00079-f002:**
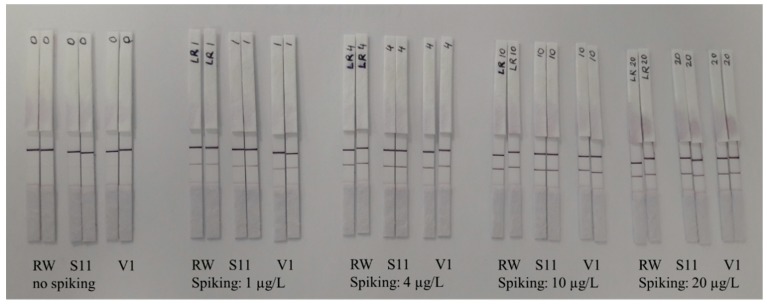
LFIA detection of microcystin-LR with concentrations ranging between 1–20 µg/L from spiked water samples. RW = Reagent water; S11= Alasenjärvi water; V1 = Rusutjärvi water.

**Figure 3 biosensors-09-00079-f003:**
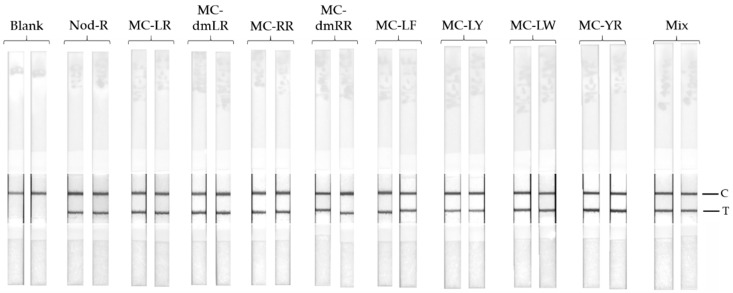
Toxin specificity of the LFIA for nodularin-R(Nod-R), nine different microcystins and a mixture of all toxins. The concentration of each toxin was set to 4 µg/L.

**Figure 4 biosensors-09-00079-f004:**
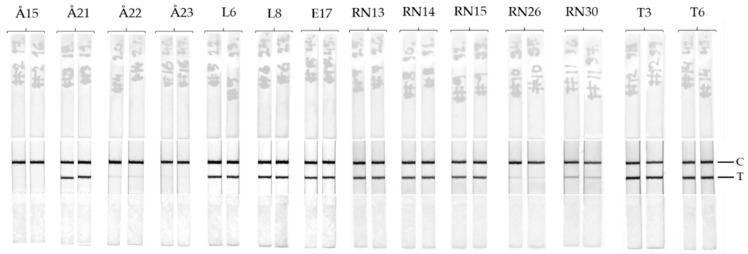
Lateral flow immunoassay with environmental samples. Surface water samples collected from various locations in southwest Finland and Estonia (14) were analyzed with the LFIA. The detailed sample information and analyzed cyanotoxin concentrations with reference methods are presented in Table 1.

**Figure 5 biosensors-09-00079-f005:**
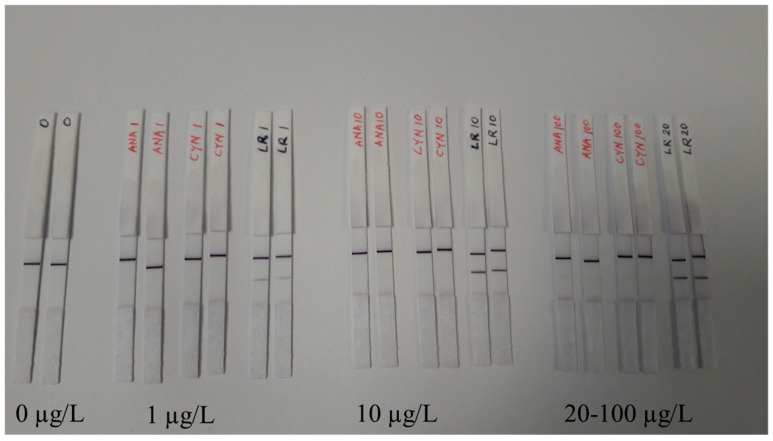
Test of possible interferences of the LFIA in the presence of other water soluble toxins. No false positive result was observed in the LFIA in the presence of 100 μg/L of anatoxin-a (ANA) or cylindrospermopsin (CYN). Blank result and positive result with microcystin-LR (1–20 μg/L) are compared.

**Table 1 biosensors-09-00079-t001:** Raw environmental samples used for the evaluation of the developed lateral flow strips assay. The corresponding toxin concentration as microcystin-LR equivalent by TRF-IC assay (raw water) and by reference LC-MS method (intracellular toxin from extracted cells) are shown.

Sample	Sample Location	Date	TRF-IC Assay ^1^ (µg/L)	LC-MS ^2^ (µg/L) Adapted from Hautala et al. [27]
			toxin content from raw water	intracellular toxin content from extracted cells
Å15	Vandö kanal, Finström, Åland Islands, Finland	28.07.2009	0	0
Å21	Nåtö hemviken, Nåtö Island, Åland Islands, Finland	30.07.2009	10.1	8.6
Å22	Nåtö vägbank (sea), Åland Islands, Finland	29.07.2009	0.9	1.5
Å23	Dalkarby träsk, Dalkarby, Åland Islands, Finland	29.07.2009	0	0
L6	Littoistenjärvi, Kaarina, Finland	26.08.2009	3.6	5.2
L8	Littoistenjärvi, Kaarina, Finland	11.09.2009	4.2	3.7
E17	Lake Harku, Estonia	18.08.2009	4.2	1.97
RN13	Hauninen reservoir, Raisio, Finland	9.06.2009	5.2	11.9
RN14	Hauninen reservoir, Raisio, Finland	16.06.2009	9.4	23.6
RN15	Hauninen reservoir, Raisio, Finland	23.09.2009	9.1	21.7
RN26	Hauninen reservoir, Raisio, Finland	1.09.2009	0.7	0.83
RN30	Hauninen reservoir, Raisio, Finland	29.09.2009	1.8	1.9
T3	Savojärvi, Pöytyä, Finland	7.08.2009	28.4	40.9
T6	Maaria reservoir, Turku, Finland	24.08.2009	39.8	49.4

^1^ Time-resolved fluoroscence based immunocomplex assay by Akter et al. 2016 [25]. ^2^ LC-MS results are adapted from Hautala et al. 2013 [27].

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
