# Peer review of "Noncompetitive Chromogenic Lateral-Flow Immunoassay for Simultaneous Detection of Microcystins and Nodularin"

_biosensors, 2019, doi:10.3390/bios9020079_

Reviewer 1 Report

The authors present a new lateral flow immunoassay for the detection of hepatoxins present in enviromental water.

Some major comments:

- I noticed that the colorimetric reaction is held for 2 h! In my knowledge, the enzymatic reaction of ALP is usually completed within 30 min. The authors should test the LFIA in different incubation times of the last step upon the substrate addition and find the optimum reaction time.

- I did not understand why the authors use a second Ab for the detectin, the Adda Mab. please explain in the text.

- The title of paragraphs 2.8 and 3.2, as well as the term at the discussion part should be changed. This experiment does not refers to specificity. Alternative title may be: The applicability of the LFIA in the detection of different hepatotoxins.

- In real sample analysis, how did the authors quantitated the samples? If so, extra details should be provided, along with a calibration graph.

- I also noticed in Table 1, that for some samples there is not a good agreement in the results between the three different methods. So the authors should change their statement at the discussion part that all three methods are in good agreement (line 309). Also, at lines 297-300 the authors report that "The use of real samples where the toxin concentration is unknown 299 can create problems for the accurate detection of cyanotoxins mainly resulting from cross-reacting 300 compounds causing interferences and unspecific matrix effects". For that reason accuracy data should be provided by calculating the recovery of spiking samples.

- Also, reproducibility darta of LFIA are missing.

- The images of the LFIAs is preferred to be provided in color-format rather than in greyscale mode.

And some minor comments:

- The paragraph 2.5 is missing

- In figure 1b at step 5 indicate which strip gives a positive and which strip gives a negative result.

- Please, at paragraph 2.4 add more information about the specificity of the used anti-IC scFv Ab.

Author Response

Dear Reviewer,

We thank you very much for your comments. We believe, your comments helps us to improve the manuscript. We tried to address all of your comments. In addition we also revised the manuscript to improve its overall quality.

Thanks and Regards,

Sultana Akter

Reviewer 2 Report

 The paper entitled: “Noncompetitive chromogenic lateral-flow immunoassay for simultaneous detection of microcystins and nodularins” describes the two-sites sandwich assay based on lateral flow assay for the rapid detection of MCs and Nods for the assessment of water quality and safety with a LOD below 4 ug/L, without the need of labelling conjugate and just using the alkaline phosphatase activity.

The concept of the paper is very attractive and it is clear and well written. I recommend the publication after minor revision.

1. The authors described in materials and methods the prepared the scFv-AP conjugate. However, there was not any result showing this resulting complex. I recommend the authors to perform at least an electrophoresis gel, demonstrating the conjugation and purity of the complex.

2. Why did the authors immobilize the HAMA antibody instead of immobilizing the monoclonal anti-Adda antibody directly on the membrane? 

3. Did the authors optimize concentration and buffer conditions for the immobilization of capture antibodies on the membrane?

4. The authors demonstrated the specificity of the assay for 9 different MCs and Nods. However, the authors should test other toxins that are not related to MCs and Nods and can be found in the water, in order to confirm the specificity of the assay.

Author Response

Dear Reviewer,

We thank you very much for your comments. We believe, your comments helped us to improve the manuscript. We tried to address all of your comments. In addition we also revised the manuscript to improve its overall quality.

Thanks and Regards,

Sultana Akter

Round  2

Reviewer 1 Report

Accept in its present form. The authors confronted well with the reviewers' instructions.